# Human Gut Microbiome: A Connecting Organ Between Nutrition, Metabolism, and Health

**DOI:** 10.3390/ijms26094112

**Published:** 2025-04-26

**Authors:** Sandra Valencia, Martha Zuluaga, María Cristina Florian Pérez, Kevin Fernando Montoya-Quintero, Mariana S. Candamil-Cortés, Sebastian Robledo

**Affiliations:** 1Centro de Bioinformática y Biología Computacional de Colombia—BIOS, Grupo de Investigación—BIOS, Parque los Yarumos, Manizales 170002, Colombia; mariana.candamil@bios.co; 2Departamento de Ciencias Básicas de la Salud, Facultad de Ciencias para la Salud, Universidad de Caldas, Calle 65 # 26-10, Manizales 170004, Colombia; mcflorian526@gmail.com (M.C.F.P.); kefemoqu@gmail.com (K.F.M.-Q.); 3Dirección Académica, Universidad Nacional de Colombia, Sede De La Paz, Km 9 Valledupar—La Paz, Cesar 202010, Colombia; srobledog@unal.edu.co

**Keywords:** gut microbiome, nutrient assimilation, microbiome modulation, microbiome biochemistry

## Abstract

The gut microbiome plays a vital role in human health, functioning as a metabolic organ that influences nutrient absorption and overall well-being. With growing evidence that dietary interventions can modulate the microbiome and improve health, this review examines whether healthcare systems should prioritize personalized microbiome-targeted therapies, such as probiotics, prebiotics, and microbiota transplants, over traditional pharmaceutical treatments for chronic diseases like obesity, diabetes, cardiovascular risk, and inflammatory conditions. A systematic review using Web of Science and Scopus databases was conducted, followed by a scientometric analysis. Key metabolic pathways, such as dietary fiber fermentation and short-chain fatty acid production, were explored, focusing on their impact on lipid and glucose metabolism. The interactions between microbial metabolites and the immune system were also investigated. Dietary interventions, including increased fiber and probiotic intake, show potential for addressing dysbiosis linked to conditions, such as type 2 diabetes, obesity, and autoimmune diseases. The review emphasizes the need to incorporate microbiome modulation strategies into clinical practice and research, calling for a multidisciplinary approach that integrates nutrition, microbiology, and biochemistry to better understand the gut microbiome’s complex role in health.

## 1. Introduction

The gut microbiome is an essential component of human health and has a function similar to a metabolic organ that influences nutrient absorption and overall well-being and co-evolves with the host. Understanding this interaction is crucial for comprehending its role in health and disease [1,2].

Trillions of microorganisms colonize various human body sites, including the skin, mouth, gastrointestinal tract, and other systems, forming a complex ecological community that establishes a symbiotic relationship with the host and helps maintain physiological homeostasis [2]. However, a significant proportion of these microorganisms cannot be cultivated in vitro due to the difficulty of replicating the natural environmental conditions in a laboratory setting. This limitation, caused by inadequate microbial isolation media or a lack of understanding of the required conditions, restricts the complete knowledge of this interaction. It is estimated that 99% of microbial communities are inaccessible through traditional microbiological isolation methods, such as artificial culture media and biochemical characterization [2,3].

Significant technological advances in omics platforms, such as metagenomics, have enabled a better understanding of microbial composition, health–disease relationships, new therapeutic opportunities, and personalized medicine. These approaches involve sequencing millions of DNA fragments simultaneously, providing detailed information about genome structure, genetic variations, gene activity, and gene behavior changes, thereby allowing strain-level insights [4]. Additionally, metabolomic studies facilitate phenotyping of the host–microorganism interaction.

This molecular approach has provided increasing evidence regarding the impact of dietary interventions on modulating the microbiome, opening new possibilities for improving health and treating various chronic diseases [5,6,7,8]. The gut microbiome is now widely studied and considered a key factor in host health, with research into dysbiosis mainly focusing on chronic disease contexts [9]. For instance, seminal articles discuss specific diseases such as type 2 diabetes, obesity, and inflammatory and immune disorders.

This review examines whether healthcare systems should prioritize personalized therapies targeting the microbiome, such as the use of probiotics, prebiotics, and microbiota transplants, over traditional pharmaceutical treatments for conditions such as obesity, diabetes, cardiovascular risk, and inflammatory diseases.

To achieve this objective, a systematic review was conducted using the Web of Science (WoS) and Scopus databases, followed by a scientometric analysis. The study explored key metabolic pathways involving the gut microbiome, including dietary fiber fermentation and short-chain fatty acid production, focusing on their impact on lipid and glycemic metabolism. The interactions between microbial metabolites and the immune system were also investigated, highlighting the potential of dietary interventions, such as increased fiber and probiotic consumption, in treating dysbiosis associated with diseases, such as type 2 diabetes, obesity, and autoimmune disorders.

## 2. Methodology

The methodology consisted of four stages. The first stage involved a literature search conducted in the WoS and Scopus databases, using the following equation: (TITLE (“human microbiome” OR microbiota OR “intestinal microbiome” OR prebiotics OR probiotics OR posbiotics) AND TITLE (biochemistry OR metabolism OR metabolomics)) AND (LIMIT-TO (DOCTYPE, “ar”) OR LIMIT-TO (DOCTYPE, “re”)). The search was confined to the fields of Medicine; Biochemistry, Genetics and Molecular Biology; Immunology and Microbiology; Pharmacology, Toxicology and Pharmaceutics; and Chemistry, covering the period from 1972 to 5 December 2024. The second stage pertained to data cleaning, where the search results from each database and the cited references of each article were downloaded and merged into a single dataset, with duplicate records removed. The third stage involved a scientometric analysis of the data, examining the dynamics of scientific production, the most prolific authors and journals, and collaborations among countries and authors. Finally, an automated prioritization of the articles was performed using the Tree of Science (ToS) algorithm [10].

Thus, this article is divided into four sections. First, the scientometric analysis; second, the description of the human microbiome and interaction with the host; third, the overall biochemical contribution of the microbiome to human health; and fourth, emerging therapies.

## 3. Results

The search yielded 1669 results in WoS and 1971 in Scopus. After data cleaning and removing duplicates, 2411 records remained (Figure 1). From this dataset, the metadata of the articles were extracted to conduct the scientometric analysis and document prioritization using the ToS algorithm.

### 3.1. Scientometric Analysis

Regarding the dynamics of scientific productivity in this area of knowledge, Figure 2 shows that it is a topic of growing interest within the scientific community (24.46%), with publication peaks reached in recent years. Additionally, it can be observed that the inflection point for exponential growth is around the year 2013, which coincides with the highest citation peak. During 2012, the most cited article was by Tremaroli and Bäckhed [11], which is a literature review on the effect of microbiota interactions on the host. This article highlights the relationship between gut microbiota and the development of chronic diseases and obesity. Citation peaks were also observed in 2016 and 2018. During these years, the most cited articles were by Morrison and Preston [12] and Lu et al. [13]; these are literature reviews related to the impact of microbiota on metabolism.

Table 1 lists the 10 countries with the highest productivity in this area of knowledge, with China leading the research in this field. Approximately 55% of all scientific output comes from China. In contrast, the United States follows in productivity with about 10% of the contributions. However, the citation rate between China and the United States is comparable. The most cited recent article (from the last three years) from China is by Xiao et al. [14]. The main finding of this study is that modulating the gut microbiome through fecal microbiota transplantation can improve chronic cerebral hypoperfusion-induced gut dysbiosis, cognitive impairment, and depression-like behaviors. This effect seems to be achieved by increasing the relative abundance of short-chain fatty acid-producing bacteria.

Regarding collaborations between countries, Figure 3a shows that there is a dominant group of countries in Community 1, while Communities 2, 3, and 4 have a similar size. However, Community 5 is the smallest, primarily composed of Latin American countries. Figure 3b reveals a significant increase in variation from 2021 onwards, indicating that the number of new collaborations (links) is growing at a higher rate than the number of newly added countries (nodes). This suggests that the academic community in this field is strengthening and consolidating. Finally, Figure 3c highlights that the United States and China have the highest number of publications and the strongest international collaborations. Recent studies in both countries have focused on how alterations in gut microbiota and bile acid metabolism affect host physiology [15,16]. Additionally, there is a high frequency of collaborations between the United Kingdom, Italy, and Luxembourg [17,18,19]. The compact structure of the network further indicates a high degree of interconnectivity, reflecting a well-established and growing research community.

Table 2 presents the journals with the highest production. The *Journal of Agricultural and Food Chemistry* presents the highest H-index, and the most cited article is by Duan et al. [20]. The main finding of this article is that flavonoids from whole oats have an antihyperlipidemic effect in mice with high-fat diet-induced hyperlipidemia. This effect is achieved through the regulation of the gut–liver axis involving the modulation of bile acid metabolism and gut microbiota. Moreover, the *International Journal of Biological Macromolecules* exhibits the highest impact factor, and the most cited article in this topic is by Chen et al. [21]. The main finding of this article is that polysaccharides from *Ganoderma lucidum* can exert antidiabetic effects in rats with a high-fat diet and streptozotocin-induced type 2 diabetes. This fungus achieves these effects by restoring the altered gut microbiota and normalizing the metabolism of various metabolites in the host.

Figure 4a shows that nine journal communities were identified in the overall dataset. Among them, Communities 1 and 2 are the largest, indicating that these research areas have the highest concentration of journals. Figure 4b reveals a stabilization in the proportion of new journals compared to new citation links from 2015 onwards, confirming the consolidation of this research field. Finally, Figure 4c presents the citation network among journals, highlighting three distinct communities represented in green, orange, and purple. The thickness of the links indicates the strength of the connections, while node size corresponds to degree centrality, reflecting how many connections each journal has with others. The purple cluster is related to pharmacology, and the orange group focuses on the influence of the environment on the microbiome and microbiota [22,23]. The orange cluster also examines the effects of external factors, such as environmental pollutants and dietary interventions, on the gut microbiota and their association with various health outcomes [24,25]. Lastly, the green community explores the impact of the microbiome on nutrition, emphasizing the therapeutic potential of mushroom-derived polysaccharides in enhancing gut health and treating inflammatory diseases [26,27]. These clusters illustrate the interconnected research areas addressing the gut microbiota’s role in health and disease. The nodes and links through time analysis also show that, since 2014, citation links between journals have increased at a higher rate than the addition of new journals, reinforcing the idea that the most-cited journals in this field have solidified their influence.

Table 3 presents the most prolific authors. The most recent article presented by Wang Y as a principal investigator is a review of recent research on the role of probiotics in regulating lipid metabolism disorders induced by a high-fat diet. The review sought to provide a critical perspective on the regulatory function of probiotics, including the selection criteria and general sources of probiotics with lipid-reducing capabilities, with the goal of promoting the development of functional foods containing natural probiotic strains that can improve these disorders. Additionally, the review explored how probiotics might activate the AMPK signaling pathway to regulate fat synthesis and breakdown, positively influence gut microbiota structure, enhance intestinal barrier function, and modulate systemic inflammatory responses. Furthermore, it examined how these effects are amplified along the gut–liver axis through the HMGCR/FXR/SHP signaling pathways [28].

Researcher Heping Zhang, with the highest h-index of 57 among the authors analyzed, focuses his research on the microbiome, probiotic agents, and gut microbiota [29]. In recent studies, Zhang et al. has emphasized the role of fermentation and probiotics in enhancing the nutritional, functional, and techno-functional properties of food products [30,31]. The second-highest h-index (38) belongs to Researcher Jiachao Zhang from Hainan University. Recent studies by Jiachao Zhang’s team explore the dual aspects of food processing, examining its potential both to generate harmful compounds and to improve nutritional properties [32,33].

### 3.2. Human Microbiome and Interaction with the Host

Trillions of microorganisms live in our body (a variety of bacteria, archaea, eukaryotes, and viruses), in deep symbiotic relationship with the host, an interaction that helps maintain physiological homeostasis, called microbiota [34,35,36]. The set of these microorganisms, their associated genes and secreted molecules, their interactions in a specific niche such as in the gastrointestinal tract is called the microbiome [36], which aids in the digestion of food, breaks down nutrients, and facilitates the absorption of essential vitamins and minerals. The study of the structure and function of complete nucleotide sequences isolated and analyzed from all organisms (usually microbes) is referred to as metagenomics [37].

Metagenomics refers to the use of bioinformatics tools and technologies that improve the knowledge of microorganisms in a massive sample [38]. This molecular tool allows rapid diagnosis of microorganisms without the need for culture. In other words, metagenomics is based on the genomic analysis of a matrix, which contains more than one microorganism and allows the complete composition of the sample to be revealed, thus allowing the detection of a diversity of bacteria, archaea, fungi, and protists.

Regarding the specific study of bacteria, the most used method to achieve this purpose is the sequencing of the 16S rRNA gene, which is an essential component of the 30S ribosome [39] and consists of more or less 1500 base pairs in length, allowing a good resolution for phylogenetic studies by efficiently exploring the diversity, functions, and information of taxa and species in an environment [40], even if they are not culturable. The 16S rRNA gene is common to all bacteria and archaea and has highly conserved regions, making it a useful marker gene for the use of universal primer sequences to isolate for sequencing. Scattered among the conserved regions of the gene are nine hypervariable regions (called V1 through V9), and it is these segments that allow taxonomic identification of organisms when mapping reads to a database of known 16S rRNA sequences [41,42].

Most of the research on the microbiome, including the pioneering work done by the Human Microbiome Project, has led to the health sciences no longer being able to deal only with the study of cells, tissues, organs, and functional systems of the human body and needing to focus also on the interactions of human cells with microbiomes in different parts of the body. Thus, functional metagenomics should be considered part of systems biology [43], as the interaction between these microorganisms and their host can be critical for health and disease [44]. Currently, the human microbiome is even considered to be our “ultimate organ” [45].

Indeed, novel and important concepts have been contributed to describe host–microbe interactions, such as the holobiont theory or the concept of metaorganism, defined as the sum of the genetic information of the host and its microbiota. The theory is based on four generalizations: (1) All animals and plants establish symbiotic relationships with microorganisms. (2) Symbiotic microorganisms are transmitted between generations. (3) The association between host and symbionts affects the fitness of the holobiont within its environment. (4) Variation in the hologenome (the set of microorganisms that make up the microbiome of a species) may be due to changes in the genome of the host or the microbiota [46,47].

These theories are based on one trillion bacteria in the human organism, approximately the same number as somatic cells, but occupying a smaller mass (200 g in an individual of 70 kg body mass) [48]. Their configuration depends on several factors throughout life: (1) mode of delivery (vaginal or cesarean section); (2) diet during infancy (breast milk or formula) and adulthood (meat-based or vegan); and (3) use of antibiotics or antibiotic-like molecules derived from the environment [46,49]. They are localized throughout the body with a higher concentration in different parts of the organism (Figure 5), the most complex, diverse, and numerous being those associated with the digestive tract, particularly in the cecum [50].

The gut microbiome’s function is to contribute to human health by increasing the stability, resilience, or optimal functioning of the body. It has been defined by global ecological parameters such as the richness, diversity, and balance of its microbial communities [52]. An increased richness and diversity of bacterial species in the human gut can be an indicator of health [53].

Of the 1013 to 1014 intestinal microorganisms, a large percentage of the microbiota are dominated by the phyla *Firmicutes* (synonym *Bacilliota*) and *Bacteroides* (synonym *Bacteroidota*), while the phyla *Actinobacteria* (synonym *Actinomycetota*), *Proteobacteria* (synonym *Pseudomonadota*), and *Verrucomicrobiota* are found in modest, similar proportions in healthy animals [54,55], playing an important role in ensuring proper digestive functioning, and in the immune system [34], in performing a barrier effect and assist in the production of neuroimmunoendocrine modulatory molecules. The gut microbiome includes more than 1000 species and several million genes. It is important in food digestion and nutrient extraction, in modifying host immune response, in protection against infections, in drug metabolism, and in the participation and regulation of host metabolism. At the same time, the microbiome is modifiable by diet, host, and environmental factors [56].

An imbalance or loss of diversity can result in so-called “dysbiosis” which describes the altered composition of microbes, generating a cascading impact on the immune system and an advantage for the emergence and outbreak of pathogens [57].

There is a wealth of research linking microbiome dysbiosis to a particular disease; for example, gut dysbiosis is associated with the pathogenesis of both intestinal and extraintestinal disorders. Intestinal disorders include inflammatory bowel disease, irritable bowel syndrome (IBS), celiac disease, and a chronic inflammatory condition of the gastrointestinal tract called Crohn’s disease, where altered species concentration is evident (Figure 6). Extraintestinal disorders include allergy, asthma, metabolic syndrome, cardiovascular disease, and obesity [58].

It is logical to predict that missing or extra microbial taxa could affect microbial interactions and secreted metabolites, which in turn can change host metabolism and other bodily functions. Several factors contribute to these microbiome–host interactions, including ecological, epigenetic, and genetic components [59]. Indeed, the mechanisms by which microbes residing in the human body interact with metabolites to impact disease risk are beginning to be elucidated, and discoveries in this area will likely be leveraged to develop preventive and treatment strategies for complex diseases [60].

Initiatives such as MetaHIT (http://www.metahit.eu/ accessed on 5 December 2024) and the Human Microbiome Project (http://hmpdacc.org/) have described the composition and molecular functional profile of the gut microbiome.

**Figure 6 ijms-26-04112-f006:**
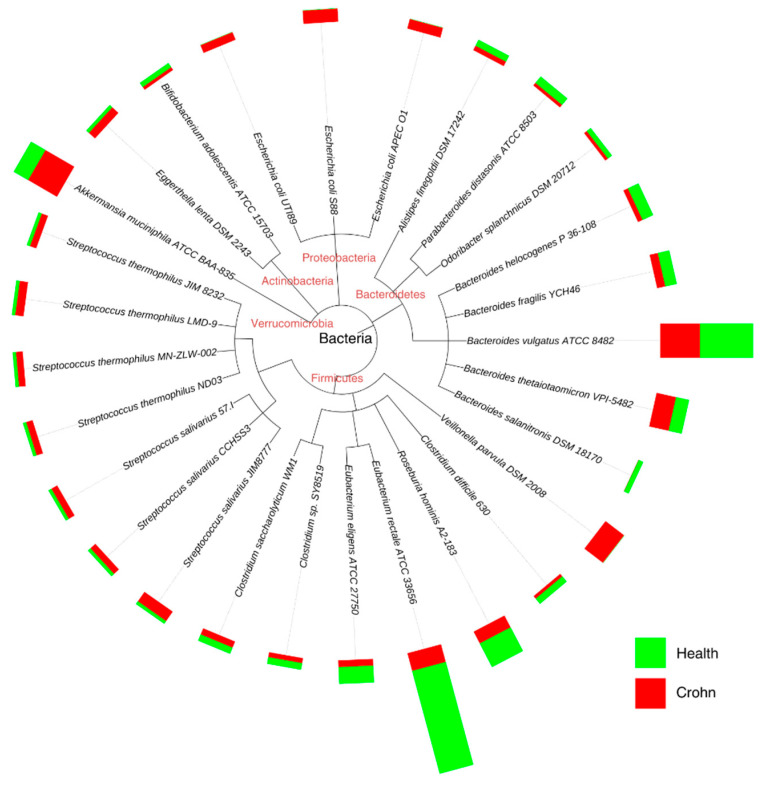
Accurate genome relative abundance estimation for closely related species in a metagenomic sample. Adapted from [61].

Beyond microbial richness and diversity, a healthy gut microbiome can be defined by the presence of classes of microbes that enhance metabolism, resilience to infection and inflammation, resistance to cancer or autoimmunity, endocrine signaling, and brain function (brain-gut axis) [62]. The microbiome may mediate these effects through the secretion of factors that modulate intestinal permeability, mucosal lining, epithelial cell function, innate and adaptive immunity, intestinal motility, and neurotransmission [63]. Indeed, there is compelling evidence supporting the role of the gut microbiota in regulating adiposity and body weight, and it has received increased attention from researchers worldwide [64].

The colon is the primary site of this fermentation, as its relatively high transit time and pH, along with low cell turnover and redox potential, present more favorable conditions for bacterial proliferation [65]. However, that does not preclude importance elsewhere. For example, in the small intestine it has been shown to regulate nutrient absorption and metabolism performed by the host [66].

High concentrations of fermentation by-products, mostly acidic, lower the pH, which creates an inhospitable environment for pathogens, although if invaders ferment, they generate toxic metabolites, damaging the epithelium and causing inflammation [56].

The gastrointestinal tract is divided into the stomach, duodenum, jejunum, ilium, and colon, whose environment is in ascending order of pH and is anaerobic from the stomach to the colon (Figure 5). The gut microbiota is established in stages throughout early life; in recent years, discoveries made using next-generation sequencing techniques in samples of placenta, amniotic fluid, meconium, and even fetal tissues have challenged the belief that the uterus is a sterile environment [67]. At birth, studies have shown that there are clear differences in the microbiome profiles of newborns born vaginally compared to those born by cesarean section [68,69]; babies born naturally are initially inoculated by bacteria typically present in the vaginal and fecal microbiome such as Lactobacillus and Prevotella spp, whereas those born by cesarean section are colonized by bacteria from the skin and environment [70]. In fact, the type of feeding from the moment of birth directly influences biodiversity; this highlights the importance of breastfeeding, the recommended source of nutrition for infants for its optimal balance of nutrients such as carbohydrates, lipids, amino acids, proteins, vitamins, and minerals indispensable for the baby’s development [71], as well as immunocompetent cells, immunoglobulins, fatty acids, polyamines, oligosaccharides, lysozyme, lactoferrin, and other glycoproteins, in addition to microorganisms and antimicrobial peptides [72,73], which help the development of innate immunity in the neonate. Likewise, oligosaccharides are important as they provide nutrients for microbial metabolism, allowing a successful symbiotic relationship between microorganism and host to occur [74,75]. In particular, regarding the microorganisms present in breast milk, species such as *Weisella, Leuconostoc, Staphylococcus, Streptococcus, Lactococcus, Veillonella, Leptotrichia,* and *Prevotella* have been reported [76] as supporting the argument of human breast milk as an optimal source of nutrition [77], being species necessary for the maturation of the neonatal gut. Different studies suggest that the microorganisms present in the mammary gland come from the intestine via intestinal monocytes that move during pregnancy and lactation [71]. A baby consumes approximately 1x10^5^ to 1x10^7^ bacteria daily, which have been found in the intestinal microbiota of the newborn and coincide with the time of weaning where little by little a decrease of the microbiota in the mammary gland is evidenced, thanks to apoptosis resulting in mammary involution [76].

The abundance of phyla in the infant microbiota depends on the order of appearance; however, in adults, the abundance is shifted: *Bacilliota* (~65%*), Bacteroidota (*~25%*) Actinomycetota (*~5%*), and Pseudomonadota (*~5%*)*; the members of *Pseudomonadota, Fusobacteriota, Cyanobacteriota, and Verrucomicrobiota* are present in adults but are less represented [51], although community dynamics are largely determined by host diet composition.

Many factors, such as diet, could be manipulated to alter the host gut microbiome and induce preventive and therapeutic effects [78]. This is due to the ability of the microbiome to convert unabsorbable products of metabolism into metabolites that can be digested by the human body. This ability is fundamental to optimize the digestion and absorption of nutrients such as fiber and other complex compounds that the human body cannot digest on its own, transforming them biochemically into other molecules that the intestine can absorb and use efficiently. This function not only maximizes the obtaining of energy from food but also influences the regulation of the immune system and disease prevention [79].

### 3.3. Biochemical Contribution of the Microbiome to Human Health

To meet the basic nutritional requirements in the human body, five types of fundamental nutrients are used, the so-called macronutrients, which are three (carbohydrates, lipids, proteins), and the two micronutrients (vitamins and minerals). Each of them plays an important biochemical role in metabolism for the maintenance of the body’s homeostasis. There is a great diversity of foods that contain them, which after undergoing processes of digestion, absorption, and transport from the small intestine to each of the tissues, mainly the liver, exert cellular metabolisms depending on the needs of the tissues [80]. However, there is part of the food that is not digestible and from the small intestine it passes to the colon where it becomes a nutritious broth for the microbiome and these microorganisms, by presenting a greater variety of enzymes, will digest and use the metabolites for their own growth, filling their niche to the maximum [81], inhibiting the proliferation of pathogens and generating bioactive metabolites easily assimilated by the human body that bind to target receptors; activate signaling cascades; and modulate several metabolic pathways with local and systemic effects [82], one of the most important being the immune system [83]. However, biochemical effectiveness in maintaining health depends on two fundamental aspects: the quality of the nutrients and the microorganisms that inhabit the gastrointestinal system.


**Macronutrient metabolism**


Carbohydrates: Classified as a primary source of energy, they contain mainly starch polysaccharides, disaccharides such as sucrose, maltose, and lactose, with linkages that only a few classes of enzymes can catalyze in gastrointestinal metabolism such as salivary and pancreatic α-amylase, disaccharidases located at the brush border of the enterocyte, such as lactase, maltase, sucrase, and trehalase providing monosaccharide units of glucose, galactose, and fructose, ready for absorption and utilization. However, polysaccharides where their units are linked differently, such as inulin, corn fiber, polydextrose, and citrus pectin, cannot be hydrolyzed by these enzymes and become non-digestible dietary fiber [84], which microorganisms can digest.

Bacteroidota that are commonly recognized as primary polysaccharide degraders due to enzymatic activity are *Bacteroides fragilis, Bacteroides intestinalis, Bacteroides thetaiotaomicron,* and *Bacteroides xylanisolvens* [85]; they possess the enzyme pool (CAZymes) and are closely associated with the degradation of complex glycans, and these enzymes are divided into two main groups, glycoside hydrolases (GH) and polysaccharide lyases (PL), although carbohydrate esterases (CE) also play important roles on some substrates [86,87,88]. A prime example is *thetaiotaomicron*, which is part of the large community of indigenous microorganisms that help shape gut physiology and a dominant member of the normal distal gut microbiota as it possesses 260 glycoside hydrolases alone in its genome [89]. *Bacilliota* ferment simpler polysaccharides and short-chain fatty acids. Among these, bacteria of the genus Clostridium (*C. butyricum* and *C. sporogenes*), the genus Lachnospira (*L. multipara*), and Eubacterium (*E. rectale*), as well as the genus Roseburia (*R. intestinalis*) and Faecalibacterium (*F. prausnitzii*), are notable for metabolizing resistant starch into butyrate, whereas bacteria of the genus Blautia (*B. obeum*) produce acetate and propionate. These microorganisms play a key role in intestinal homeostasis by reinforcing the epithelial barrier and modulating inflammation [90].

Dietary fiber and whole grains contain a unique blend of bioactive components including resistant starches, vitamins, minerals, and phytochemicals and can be divided into non-fermentable/insoluble, such as celluloses, hemicelluloses, fructans, and prebiotics [91], and fermentable/soluble that generate short-chain fatty acids (SCFA), namely acetate, propionate, and butyrate, in a 60:20:20 ratio [92], a ratio that changes depending on the quality of the fiber and microbiome, before the non-digestible matter reaches the rectum, and which are responsible for modulating several metabolic pathways that are implicated in obesity, insulin resistance, and type 2 diabetes [93]. The first two are absorbed into the portal circulation, and butyrate serves as an energy substrate for colonocytes [94]. Now, soluble fibers such as pectin are characterized by being viscous, creating an intestinal barrier that slows the absorption of nutrients such as glucose and lipids, presenting a protective factor for patients with diabetes [95].

Proteins: Amino acids (aa) are the fundamental units of proteins. Proteins are hydrolyzed in the stomach and later in the small intestine, where they continue to break down into free amino acids, dipeptides, or tripeptides, which are then absorbed by enterocytes. Some of these amino acids become the primary energy source for enterocytes and are also available to the gut microbiota of mammals [96]. Studies show that, in the case of low-protein diets, the microbiota becomes a crucial source of amino acids, particularly through the action of Firmicutes, which the host can use for energy or to build structural tissues [97]. The intestinal microbiota can synthesize several nutritionally essential amino acids de novo, serving as a potential regulatory factor since microbial amino acids contribute to the host’s amino acid homeostasis [98]. Alterations in the composition and quantity of amino acids can affect the bacterial communities that metabolize them, as well as modulate macrophages and dendritic cells through receptors such as toll-like (TLR), autoinducer-2 (AI-2), and NOD-like receptors (NLR) [96]. Additionally, they regulate the gut-microbiome–immune axis through receptors like the aryl hydrocarbon receptor (AhR) and serotonin/5-hydroxytryptamine (5-HT), among other signaling pathways, playing essential roles in regulating mucosal immunity and the microbiota directly or indirectly, thus contributing to intestinal homeostasis [99]. Recent studies focus on the effects of specific functional amino acids on the gut-microbiome–immune axis, highlighting the signaling pathways of tryptophan (Trp), glutamine (Gln), methionine (Met), and branched-chain amino acids (BCAAs) in the intestinal barrier and their relation to immunity through interaction with their receptors or ligands [99].

Dietary proteins not fully digested in the small intestine can reach the colon, where they are broken down by the microbiota into peptides, free amino acids, and further metabolized, generating various microbial metabolites like short-chain fatty acids (SCFAs), polyamines, hydrogen sulfide, phenol, and indole, which play crucial roles in physiological functions related to host health and disease [100].

In this context, extensive research has been conducted on how animal and plant dietary proteins influence intestinal microbiome modulation. Various studies have shown that plant proteins, particularly soy, have gained popularity for their health benefits, while animal proteins are considered higher-quality proteins [101].

Lipids: They are fundamental for providing structure to the cell membrane, storing energy, yielding nearly twice as much as the catabolism of carbohydrates or proteins, regulating essential biological functions like cell signaling and apoptosis, being a significant part of some hormones and bile salts, and acting as mediators in the central nervous system and immune response. Lipid metabolism also affects genetic transcription by influencing chromatin state and modulating protein function through post-translational modifications [102]. It is evident, then, that a lipid metabolic disorder could have physiological consequences affecting health.

Lipid digestion, particularly of triglycerides, begins in the mouth with the action of the lingual lipase, a highly specific acyl-ester hydrolase, which remains active in the stomach, preferentially hydrolyzing short-chain fatty acids at position 3, which can be absorbed in the stomach for energy directed to the liver. The remaining fatty acids pass to the small intestine, where they are primarily hydrolyzed by pancreatic lipase at position 1; the monoacylglycerols (MAG) at position 2 theoretically can be isomerized to position 1 and then absorbed, but this process is not highly favored due to the intestinal pH. After fatty acids and MAG enter enterocytes, re-esterification occurs, allowing triglyceride formation and their subsequent secretion into the lymph as chylomicrons, and finally into systemic circulation. Saturated fatty acids with a chain length of C18 or longer at positions 1 and 3 are not absorbed due to their high melting points, passing into the large intestine as calcium salts, which is one of the most common causes of constipation in children and adults [103]. In breast milk, lacteal lipase activates in the intestine through bile salts with the capacity to hydrolyze triglycerides at positions 1, 2, and 3, a critical function after birth since the exocrine activity of the pancreas is underdeveloped in newborns and only matures by the third week [104].

Intestinal microorganisms have a significant impact on the absorption, digestion, metabolism, and excretion of lipids. They can digest lipids obtained from food during digestion and synthesize new lipids important to our health [105]. Host or dietary lipid molecules participate in energy metabolism and in maintaining the structural integrity of cell membranes. In particular, gut microbes can convert these lipids into bioactive signaling molecules through their biotransformation pathways and short and long-chain fatty acid synthesis. Within the Firmicutes phylum, the genera Faecalibacterium (*F. prausnitzii*), Clostridium (*C. butyricum, C. sporogenes*), and Eubacterium (*E. rectale*) ferment carbohydrates and amino acids, primarily producing butyrate. Within the Bacteroidetes phylum, the genus Bacteroides (*B. thetaiotaomicron*, *B. fragilis*) degrades complex polysaccharides, generating acetate and propionate. Lastly, within the Actinobacteria phylum, the genus Bifidobacterium (*B. longum, B. adolescentis*) ferments oligosaccharides and predominantly produces acetate and lactate [106].


**Effect on human health**


The gut microbiome plays a crucial role in human health by facilitating the digestion and metabolism of nutrients that the body cannot process on its own. Through various biochemical interactions, these bacteria generate bioactive compounds such as SCFAs (primarily acetate, propionate, and butyrate), which are the end products of this process and significantly impact human health [107]. Butyrate becomes the main energy source for colonocytes. Propionate mainly targets the liver, while acetate is released into the peripheral circulation. A proper balance between acetate and propionate is associated with positive effects on metabolism [108].

The following describes the effects of SCFAs in stimulating different receptors across various organs in the body. SCFAs are recognized as essential energy sources for the host and act as signaling molecules through G protein-coupled receptors, such as FFAR2, FFAR3, OLFR78, and GPR109A. They also regulate gene expression by inhibiting histone deacetylase (HDAC) [109]. Regarding their effects on the endocrine system, SCFAs influence the regulation of satiety-inducing hormones such as peptide YY (PYY) and glucagon-like peptide-1 (GLP-1) and regulate serotonin synthesis in enteroendocrine cells (Figure 7) [110].

A crucial impact of SCFAs occurs in adipocytes and muscle tissue, where they significantly contribute to energy metabolism and the regulation of metabolic homeostasis. In adipocytes, SCFAs activate G protein-coupled receptors (such as FFAR2 and FFAR3), which not only promote lipolysis and reduce fat accumulation but also enhance the release of hormones like PYY and GLP-1 that regulate appetite and improve insulin sensitivity. This action helps lower blood insulin levels, essential in preventing insulin resistance and obesity. Additionally, SCFAs inhibit adipogenic differentiation, limiting new fat tissue formation and contributing to better metabolic balance [111].

In muscle tissue, SCFAs improve fatty acid oxidation efficiency and glucose metabolism, promoting glycogen synthesis and facilitating glucose uptake in response to insulin. This not only optimizes muscle performance but also contributes to better glycemic control, which is crucial in preventing metabolic disorders like type 2 diabetes. Furthermore, FFAR3 activation in muscles is associated with increased energy expenditure and thermogenesis, promoting greater fatty acid use as an energy source and reducing body fat accumulation [112].

In the liver, SCFAs play a crucial role in lipid and carbohydrate metabolism regulation, as well as in modulating insulin sensitivity. Propionate and acetate, two of the most studied SCFAs, are transported to the liver after their production in the colon by the gut microbiota. Once in the liver, propionate acts as a key substrate in gluconeogenesis, the process by which the liver generates glucose, and also contributes to reducing lipogenesis by inhibiting fatty acid synthesis. Additionally, propionate has anti-inflammatory effects, which are essential in counteracting chronic inflammation associated with various metabolic diseases [113]. On the other hand, acetate, while it may stimulate lipid synthesis in the liver when in excess, also has a protective role when production is balanced, helping maintain an appropriate balance between lipid production and lipids’ utilization as an energy source. This balance is fundamental to preventing hepatic fat accumulation, a risk factor for developing non-alcoholic fatty liver disease. Furthermore, SCFAs modulate gene expression in the liver through HDAC inhibition, promoting the expression of genes involved in glucose and lipid metabolism control and the inflammatory response. Overall, SCFAs in the liver contribute to improved insulin sensitivity, glucose level regulation, and prevention of metabolic liver diseases.

In the immune system, SCFAs play a fundamental role as modulators of immune response and inflammation. One of their most important functions is inhibiting histone deacetylase (HDAC), allowing for the epigenetic regulation of gene expression in immune cells. This promotes the activation of anti-inflammatory genes and reduces the expression of pro-inflammatory genes, a key process for maintaining immune tolerance in the gut and other body parts, thereby avoiding an excessive inflammatory response that could lead to autoimmune or inflammatory diseases [114]. SCFAs also act on G protein-coupled receptors like FFAR2 and FFAR3, which are found in immune cells such as macrophages, neutrophils, and dendritic cells. Activation of these receptors promotes the production of anti-inflammatory cytokines, such as interleukin-10 (IL-10), and reduces the release of pro-inflammatory cytokines, like tumor necrosis factor-alpha (TNF-α) and interleukin-6 (IL-6). This modulation is crucial for regulating the balance between pro- and anti-inflammatory immune responses [115].

In chronic inflammatory diseases, SCFAs also play a protective role. For example, butyrate strengthens the intestinal barrier by promoting the integrity of tight junctions between epithelial cells, preventing the translocation of bacteria and their products (such as lipopolysaccharides, LPS) into the bloodstream, thereby reducing systemic immune activation. This function is key to preventing chronic inflammation and diseases related to gut dysbiosis, such as inflammatory bowel disease (IBD) and metabolic syndrome [116]. SCFAs also influence the differentiation and function of regulatory T cells (Tregs), a type of immune cell that plays a crucial role in suppressing excessive immune responses and preventing autoimmunity. Butyrate, in particular, is known to promote the expansion of Tregs, contributing to the maintenance of immune tolerance and reducing inflammation throughout the body. Non-ionized SCFAs can inhibit the growth of certain pathogenic bacteria, as they can penetrate bacterial membranes, dissociate in the cytoplasm, reduce intracellular pH, and inhibit the growth of susceptible bacteria, including antibiotic-resistant *Enterobacteriaceae* [117]. In summary, SCFAs have a profound impact on the immune system, helping regulate inflammation, strengthen the intestinal barrier, and maintain the immune balance necessary to prevent inflammatory and autoimmune diseases [118]. Some potentially pathogenic bacteria may include within the phylum Pseudomonadota (Proteobacteria), *Escherichia coli* (AIEC), *Salmonella enterica*, and *Helicobacter pylori*, which can promote bacterial translocation and immune response. Within the phylum Bacteroidota (Bacteroidetes), *Bacteroides vulgatus* and toxigenic *B. fragilis* can generate inflammatory metabolites. Within the phylum Bacillota (Firmicutes), *Clostridium difficile* and *C. perfringens* produce toxins that damage the intestinal epithelium.

The gut microbiome also influences the bidirectional communication between the gut and brain, known as the gut–brain axis. Through SCFA production, the microbiome activates G protein-coupled receptors (such as FFAR2, FFAR3, GPR109A, and OLFR78) present in the nervous and immune systems. These SCFAs modulate neuronal functions, regulate cytokine production, activate microglia, and protect the blood–brain barrier, thereby reducing brain inflammation and promoting central nervous system homeostasis [119]. They also regulate neurotransmitters such as GABA, serotonin, dopamine, and norepinephrine, influencing mood, satiety, stress, and vagus nerve activity, suggesting a connection between gut dysbiosis and disorders like depression and anxiety [120].

The gut–lung axis represents another pathway of bidirectional communication where changes in the gut impact lung function and vice versa. In acute respiratory distress syndrome (ARDS), systemic inflammation can increase intestinal permeability, allowing bacterial products like lipopolysaccharides (LPS) to enter the bloodstream, thereby exacerbating lung inflammation. Additionally, gut dysbiosis reduces SCFA production, and these metabolites typically have anti-inflammatory effects that, under normal conditions, could help mitigate lung inflammation [121].

Regarding cardiovascular diseases, especially those affecting the heart, there are various metabolic and immune pathways connecting them to the gut microbiome. An imbalanced microbiome (dysbiosis) can contribute to the development of heart diseases through the production of harmful metabolites like trimethylamine-N-oxide (TMAO), which is associated with an increased risk of atherosclerosis, inflammation, and adverse cardiovascular events. Systemic inflammation caused by a compromised intestinal barrier allows bacterial products, such as LPS, to pass into the bloodstream, inducing chronic inflammatory responses that harm the cardiovascular system. Dysbiosis also disrupts SCFA production, which normally has anti-inflammatory and protective effects on the vascular endothelium and blood pressure [122].

Additionally, the gut microbiome affects the bioavailability of essential vitamins (A, B, C, D, E, and K) and minerals (such as calcium, iron, zinc, magnesium, and phosphorus), which are crucial for cellular metabolism, energy production, and blood clotting. The microbiome acts as a protective barrier by competing with pathogens for nutrients and space in the gastrointestinal tract. However, antibiotic treatment can eliminate beneficial bacteria, increasing nutrient availability for pathogen colonization [123].

### 3.4. Emerging Therapies

Modulation of the gut microbiota can be achieved through emerging therapies with promising strategies to restore or modify microbiome composition and, therefore, overall health. Microbiome therapeutics represent an innovative approach to treating various conditions associated with gut dysbiosis through microbiome engineering, using additive, subtractive, and modulatory strategies. Emerging approaches, including the use of native or modified microbes, selective antibiotics, bacteriophages, and bacteriocins, offer promising solutions for diseases ranging from intestinal infections to metabolic and neurological disorders.

Additive therapy aims to add beneficial microbes to the gut microbiome to correct imbalances or enhance specific functions through the use of probiotics or symbiotics, which can be native microorganisms like *Lactobacillus* and *Bifidobacterium* or genetically modified. Native microbes are used to restore microbial balance in cases of dysbiosis, reducing pathogenic bacteria, improving digestion, and strengthening immune function [124]. In the treatment of irritable bowel syndrome (IBS), additive therapy has proven effective through the use of autoprobiotics (autochthonous bacteria isolated from patients and cultured in the lab). A pilot clinical trial showed that these personalized probiotics contributed to the disappearance of dyspeptic symptoms and induced positive changes in the gut microbiome, such as an increase in *Coprococcus* and *Blautia* abundance and a decrease in *Paraprevotella*. Metabolic alterations, such as increased oxalic acid and decreased other metabolites, were also observed, suggesting that autoprobiotics not only restore microbial balance but also influence the body’s metabolic processes. These results highlight the potential of additive therapy, especially with personalized probiotics, for achieving stable clinical improvement in managing IBS [125].

Subtractive therapy aims to eliminate or reduce harmful bacteria in the gut through the use of antibiotics, bacteriophages, and bacteriocins. Selective antibiotics, such as rifaximin, are designed to target harmful bacteria specifically without significantly altering the healthy microbiota, proving useful in conditions like IBS. Bacteriophages are viruses that infect and destroy specific bacteria, offering a targeted approach to eliminate pathogens without affecting beneficial bacteria, which holds promise for treating antibiotic-resistant infections [28]. Bacteriocins, antimicrobial peptides produced by certain bacteria, inhibit the growth of competitors, providing a selective way to remove harmful bacteria without altering the healthy microbial balance; these are potentially beneficial in intestinal infections and microbiome modulation [126].

Modulatory therapy focuses on adjusting the composition and function of the microbiota without directly adding or removing bacteria, using strategies like diets, prebiotics, and postbiotics. Modulation through dietary changes or prebiotic supplementation (fibers that feed beneficial bacteria) promotes the growth of healthy microorganisms such as bifidobacteria and lactobacilli, thereby improving gut health and reducing pathogen proliferation [127]. Additionally, postbiotics, considered the next generation of probiotics, are extracts free of live bacteria that show modulatory effects on the gut microbiota. Examples of postbiotics include vitamin B12, vitamin K, folate, lipopolysaccharides, enzymes, and SCFAs, which have anti-obesity and anti-diabetic effects through mechanisms such as increasing energy expenditure, decreasing adipocyte formation and differentiation, and regulating gut dysbiosis. Thus, the use of postbiotics not only contributes to gut health but is also considered a promising strategy for treating metabolic disorders [128].

Emerging approaches, including the use of native or modified microbes, can be designed to perform specific functions, such as producing therapeutic metabolites, enhancing nutrient digestion, or modulating the immune system. Advances in biotechnology have enabled the creation of modified bacteria that can detect and attack pathogens, degrade toxins, or even produce drugs within the gut [129]. Selective antibiotics, bacteriophages, and bacteriocins offer promising solutions for diseases ranging from intestinal infections to metabolic and neurological disorders. Emerging therapies seek to employ more selective approaches, such as controlled broad-spectrum antibiotics [130], bacteriophages targeting specific bacteria, and bacteriocins that inhibit pathogens without affecting the rest of the microbiome. These approaches offer a more precise and safer way to treat infections and dysbiosis [131,132].

## 4. Conclusions

This review confirms that the gut microbiome plays a crucial role in health and disease, functioning as a metabolic organ that influences nutrient absorption, macronutrient metabolism, and overall homeostasis. It highlights that intrinsically beneficial microorganisms, such as *Faecalibacterium prausnitzii* and *Roseburia intestinalis*, produce short-chain fatty acids (SCFAs) with anti-inflammatory effects and strengthen the intestinal barrier. In contrast, other microorganisms, such as *Clostridioides difficile* and certain escherichia species, can induce inflammation and compromise the integrity of the intestinal epithelium.

From a metabolic perspective, different bacterial phyla play key roles in host health. Bacteroidota and Bacillota metabolize fermentable carbohydrates, thereby promoting SCFA production; Firmicutes degrade proteins, generating metabolites with both beneficial and detrimental effects; and Bacillota participate in lipid metabolism, influencing overall metabolic regulation.

Nutritional interventions, such as increasing the consumption of fiber and probiotics, may promote the growth of beneficial species while reducing the proliferation of pathogenic microorganisms, thereby helping to restore eubiosis in cases of dysbiosis associated with chronic diseases like obesity, type 2 diabetes, and autoimmune disorders. This underscores the importance of maintaining a balanced microbiome as both a therapeutic and preventive strategy for various health conditions.

## Figures and Tables

**Figure 1 ijms-26-04112-f001:**
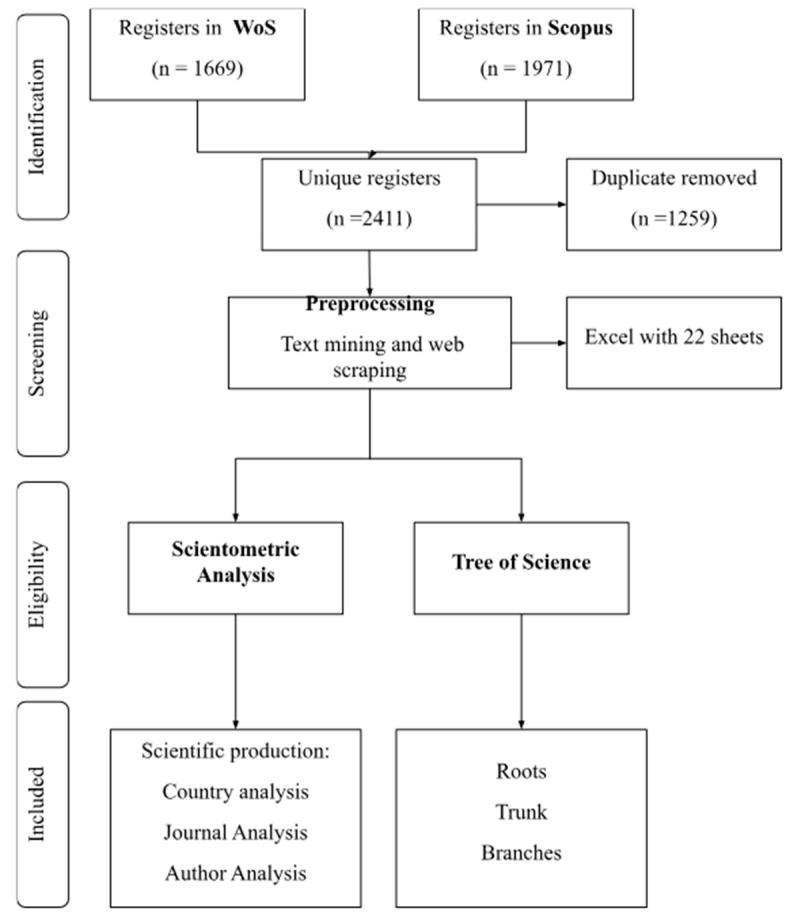
Systematic and automatized workflow for data processing and document selection.

**Figure 2 ijms-26-04112-f002:**
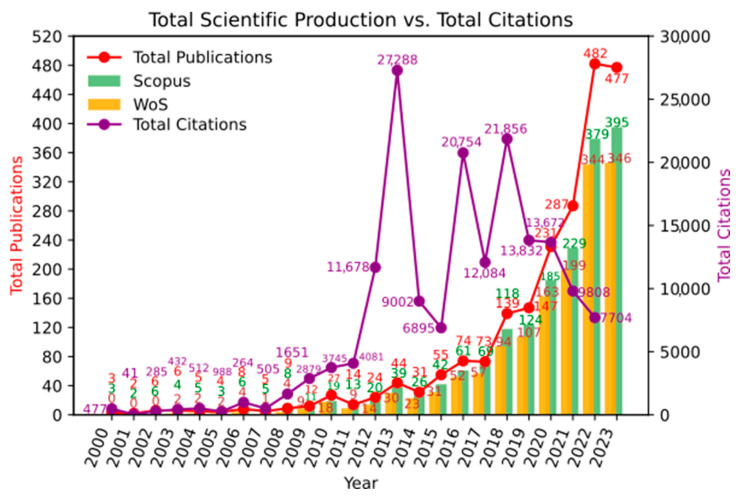
Scientific production over time.

**Figure 3 ijms-26-04112-f003:**
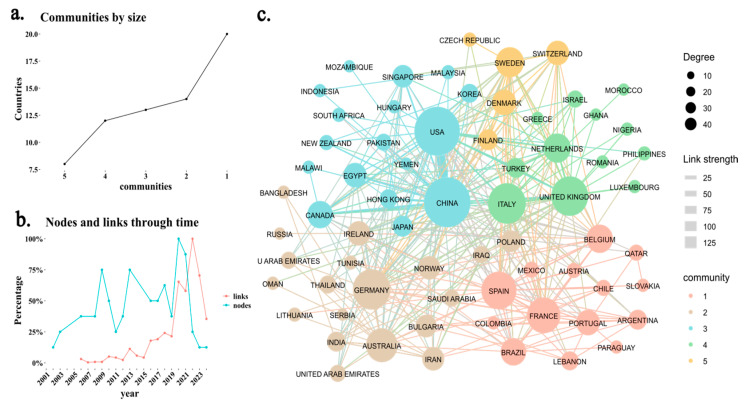
Scientific collaboration network among countries. (**a**) Distribution of scientific communities based on a clustering algorithm, identifying five distinct communities of collaborating countries. (**b**) Evolution of new scientific collaborations over time, where “nodes” represent newly added countries in the network, and “links” indicate newly established collaborations between countries in each year. (**c**) Network Scientific collaboration between countries, based on author affiliations. The size of each node represents the degree (number of connections), while the color indicates community membership. Link thickness represents the collaboration strength, with thicker links indicating stronger ties between countries.

**Figure 4 ijms-26-04112-f004:**
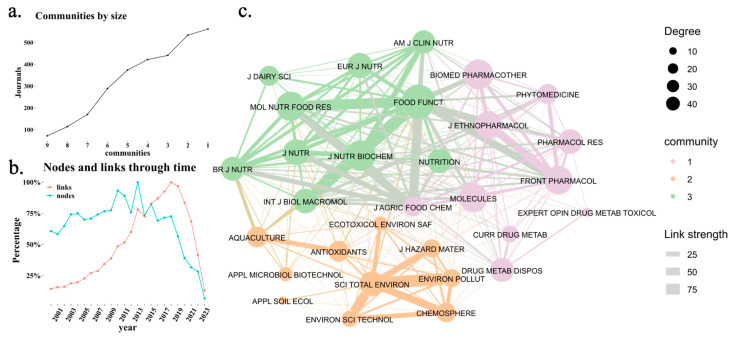
Citation network among scientific journals. (**a**) Identification of the main journal communities based on a clustering algorithm, grouping journals into thematic clusters. (**b**) Evolution of the proportion of new journals and new citation links per year, where “nodes” represent newly added journals, and “links” indicate new citation connections between journals each year. (**c**) Citation network among scientific journals, where node size represents the journal’s centrality (number of citations received), and link thickness indicates the strength of citation connections. The color of each node represents its community membership.

**Figure 5 ijms-26-04112-f005:**
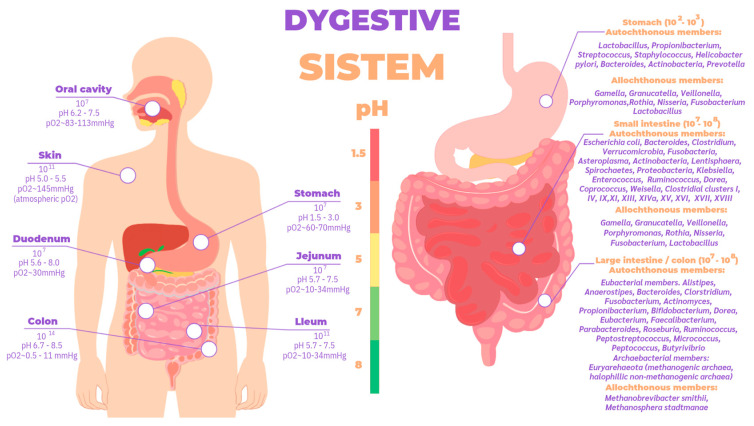
Total bacterial abundance according to different body sites and autochthonous and allochthonous Members of bacterial taxa distributed in different compartments of the TGI [9,51].

**Figure 7 ijms-26-04112-f007:**
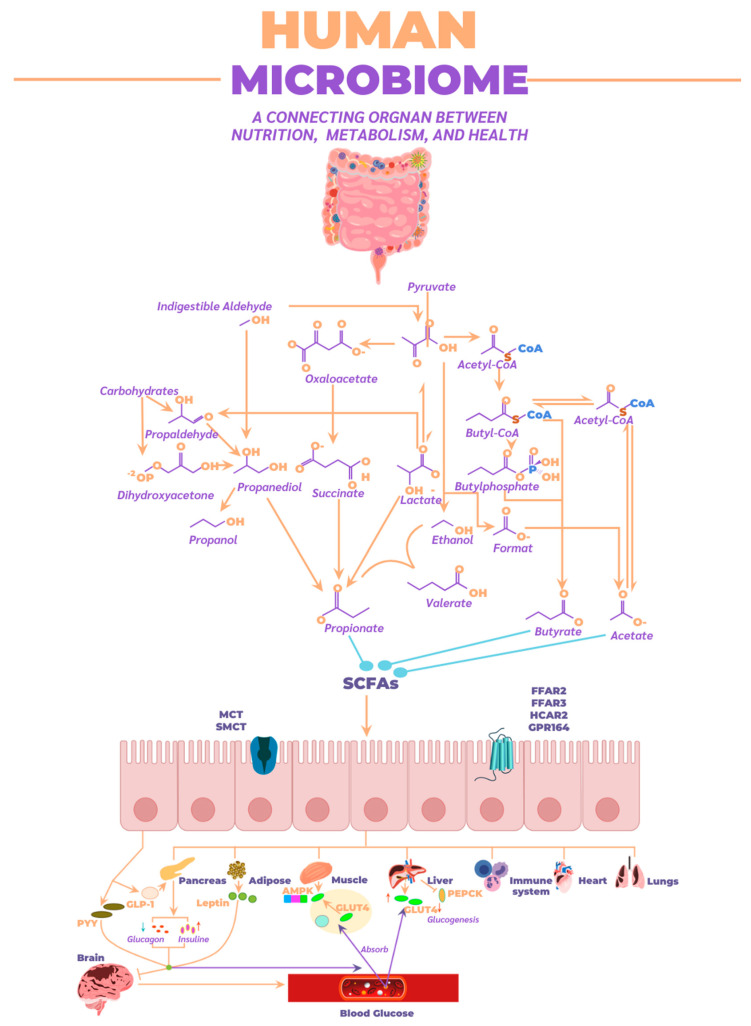
Metabolic pathways employed by the microbiome for SCFAs synthesis, along with absorption routes and their distribution throughout the human body to various organs.

**Table 1 ijms-26-04112-t001:** Academic Productivity, Citation Impact, and Quality Distribution Across Countries.

Country	Production	Citation	Q1	Q2	Q3	Q4
China	1362 (55.34%)	24,943 (23.98%)	603	83	17	17
USA	243 (9.87%)	19,000 (18.26%)	146	22	3	3
Spain	64 (2.6%)	3516 (3.38%)	43	4	3	0
Italy	62 (2.52%)	3928 (3.78%)	31	5	2	0
Japan	62 (2.52%)	2106 (2.02%)	33	12	2	1
Germany	53 (2.15%)	2541 (2.44%)	33	6	1	1
France	50 (2.03%)	4567 (4.39%)	28	6	3	1
Korea	45 (1.83%)	1116 (1.07%)	20	7	0	0
United Kingdom	45 (1.83%)	5764 (5.54%)	32	6	0	0
Canada	38 (1.54%)	1222 (1.17%)	24	2	1	0

**Table 2 ijms-26-04112-t002:** Scientific Production and Impact of Journals in the Field of Microbiology and Nutrition.

Journal	WoS	Scopus	H-Index	Impact Factor	Quantile
*Frontiers in Microbiology*	66	21	233	1.07	Q1
*Journal of Agricultural and Food Chemistry*	58	63	345	1.11	Q1
*Food and Function*	0	71	117	1.07	Q1
*Nutrients*	46	54	209	1.3	Q1
*Journal of Functional Foods*	32	37	125	0.9	Q1
*Frontiers in Pharmacology*	35	35	154	1.07	Q1
*Molecular Nutrition and Food Research*	0	37	160	1.04	Q1
*International Journal of Molecular Sciences*	29	29	269	1.18	Q1
*Frontiers in Nutrition*	25	27	77	0.83	Q2
*International Journal of Biological Macromolecules*	18	21	191	1.25	Q1

**Table 3 ijms-26-04112-t003:** Production By Principal Investigator.

No	Principal Investigator	Total Articles *	Scopus H-Index	Affiliation
1	Wang Y	148	17	University of Karachi, Karachi, Pakistan
2	Zhang Y	109	26	Beijing Solidwill Sci-Tech Co. Ltd., Beijing, China
3	Li Y	105	10	The University of Texas at Dallas, Richardson, United States
4	Liu Y	97	13	First Affiliated Hospital of Zhengzhou University, Zhengzhou, China
5	Zhang J	96	38	Hainan University, Haikou, China
6	Li X	94	16	First Affiliated Hospital of Zhengzhou University, Zhengzhou, China
7	Wang J	86	23	Dongguan University of Technology, Dongguan, China
8	Wang X	83	10	Zhejiang Provincial Hospital of Chinese Medicine, Hangzhou, China
9	Chen Y	77	37	Zhejiang University School of Medicine, Hangzhou, China
10	Zhang H	74	57	Neimenggu Agricultural University, Hohhot, China

* Total Articles is the sum of WoS and Scopus unique papers.

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
