# Peer review of "Human Gut Microbiome: A Connecting Organ Between Nutrition, Metabolism, and Health"

_ijms, 2025, doi:10.3390/ijms26094112_

Round 1
Reviewer 1 Report
Comments and Suggestions for Authors
The manuscript ijms-3519316 entitled “Human microbiome: a connecting organ between nutrition, metabolism, and health” is a systematic review on the impact of the human gut microbiome on health and disease. The authors clearly explained the adopted methodology. The content of manuscript has scientific value and adds significance to the gut microbiome field. However, same minor changes are recommended to improve the quality of the manuscript. Bellow the authors can find the reviewer endorsed modifications.
- Since the authors targeted the human gut microbiome not the human microbiome (in strict sense) the title should change to “Human gut microbiome: a connecting organ between nutrition, metabolism, and health”.
- Abstract change article to review.
- Abstract: change linked to conditions like type 2 diabetes to linked to conditions, such as type 2 diabetes
- Introduction: change habitats to various human body sites
- Introduction: change artificial media cultivation to artificial culture media
- Introduction: change microbial genome sequencing to metagenomics (in this context).
- Introduction: change type II diabetes to type 2 diabetes.
- Introduction: change This article to This review
- Introduction: change involving the microbiome to involving the gut microbiome
- Introduction: change with diseases like type 2 diabetes to with diseases, such as type 2 diabetes
- Methodology: change The methodology consists to The methodology consisted ( the study was performed, not under course).
- Methodology: the same as above, change prioritization of the articles is performed to prioritization of the articles was performed
- Methodology: What was the time frame? this information is required
- Methodology: Change to First, the scientometrics
- Results: Figure 3: Identify each graph with a letter and provide a more complete legend.
- Results: the same for Figure 4 ; Figure 4 could be divided by letters; A,B and C and a more complete legend.
- Results: check this [e.g., 22,23] (e.g.,). (e.g.,)
- Results: Cluster 2 investigates - What is the colour of this cluster?
- The approach used by the authors to comment the studies by author (Table 3) does not seem ethical. Microbiome/microbiota science is not individual. There is a principal investigator but the studies report data worked by a team. I suggest to revise this approach (as principal investigator instead of principal author).
- Page 6, where principal author is a review in which about change to principal investigator is a review about
- Page 7, Change Professor Heping Zhang, to Researcher Heping Zhang,
- Page 7, Change Dr. Zhang to Zhang et al
- Page 7, as above change Professor Jiachao Zhang to Researcher (as above)
- Page 7. Change Dr. Jiachao Zhang’s recent work to in recent studies of J. Zhang’s team
- Page 8, change microeukaryotes to eukaryotes (in the sentence you have microorganisms, so the designation microeukarayotes is not correct).
- Page 8, the definition of microbiome provide here comes from the reference of Berg et al. 2020 (Berg, G., Rybakova, D., Fischer, D. et al. Microbiome definition re-visited: old concepts and new challenges. Microbiome 8, 103 (2020). https://doi.org/10.1186/s40168-020-00875-0), please cite it instead of reference 36.
- Page 9 Verrumicrobia is presently Cite for the new phylum taxonomy the bellow reference: Oren A, Garrity GM. Valid publication of the names of forty-two phyla of prokaryotes. Int J Syst Evol Microbiol. 2021 Oct;71(10). doi: 10.1099/ijsem.0.005056.
- Page 9, Change húesped to host
- Page 10 Figure 6 is adapted from reference 61? Provide a better legend and the authorization for the use.
- Page 10, Change The intestine to The gastrointestinal tract . (The intestine does not includes the stomach!)
- Page 11, LMH in full
- Page 11, correction required for 1x105 to 1x107.
- Page 11, Also indicate here: Firmicutes (~65%), Bacteroidetes (~25%) Actinobacteria (~5%) and Proteobacteria (~5%), the members of Proteobacteria, Fusobacteria, Cyanobacteria and Verrucomicrobia " the new taxonomy
- Page 12, Bacteroidetes : indicate the new taxonomy
- Page 12, change a prime example is Bacteroides thetaiotaomicron to a prime example is thetaiotaomicron
- Page 12 Again for Firmicutes indicate the new taxonomy. Here and all over the manuscript
- Page 12, what this means generate high fermentation ?
- Page 12 change short-chain fatty acids (SCFA) acetate to short-chain fatty acids (SCFA), namely acetate
- Rewrite the sentence: Proteins: Amino acids (aa) are the functional units of proteins. These proteins ....( this is not appropriate)
- Page 12, Please revise this: These amino acids are not an energy source for bacteria, as they lack all the necessary enzymes for conversion. See the aminoacid fermentation by Clostridium species. where aminoacids are used as electron donors and acceptors (tryptophan and tyrosine can be both). See Stickland fermentation.
- Page 13, Modify this Lipids: Lipid …This is not an appropriate format for starting a sentence.
- Page 14 A more complete of legend of Figure 6 is required.
- Page 14, change Gut Microbiome to gut microbiome
- Page 15, change Immune System to immune system
- Page 15, Change Microbial metabolites can inhibit to SCFAs can inhibit
- Page 16, Againg change Gut Microbiome to gut microbiome
- Page 16, change Gut-Lung Axis to gut-lung axis
- Page 16 Change Cardiovascular Diseases to cardiovascular diseases
- Page 16, Again change Gut Microbiome to gut microbiome
- Page 16, change Gut Microbiota to gut microbiota
- Page 17, if the intention is to use of the genus names change to Bifidobacterium and Lactobacillus if the bacterial group change to bifidobacteria and lactobacilli.
- Page 17, change gut flora to gut microbiota
- References, put the reference 3 in the required format.
Author Response
Dear Reviewer,
We sincerely appreciate your valuable comments in improving our manuscript. All changes made to the document are highlighted in red.
In response to your comments, we have compiled each modification in the attached table.
Thank you once again for your time and insightful feedback.
Author Response:
Dear Reviewer,
We sincerely appreciate your valuable comments in improving our manuscript. All changes made to the document are highlighted in red.
In response to your comments, we have compiled each modification in the attached table.
|
Reviewer's comments |
Response |
|
Since the authors targeted the human gut microbiome not the human microbiome (in strict sense) the title should change to “Human gut microbiome: a connecting organ between nutrition, metabolism, and health”. |
We totally agreed with this comment. Title was changed. |
|
Abstract change article to review. |
We agreed with the comments and changes were made. |
|
Abstract: change linked to conditions like type 2 diabetes to linked to conditions, such as type 2 diabetes |
|
|
Introduction: change habitats to various human body sites |
|
|
Introduction: change artificial media cultivation to artificial culture media |
|
|
Introduction: change microbial genome sequencing to metagenomics (in this context). |
|
|
Introduction: change type II diabetes to type 2 diabetes. |
|
|
Introduction: change This article to This review |
|
|
Introduction: change involving the microbiome to involving the gut microbiome |
|
|
Introduction: change with diseases like type 2 diabetes to with diseases, such as type 2 diabetes |
|
|
Methodology: change The methodology consists to The methodology consisted ( the study was performed, not under course). |
|
|
Methodology: the same as above, change prioritization of the articles is performed to prioritization of the articles was performed |
|
|
Methodology: What was the time frame? this information is required |
The search was verified, and the missing information was added: "The search was confined to the fields of Medicine; Biochemistry, Genetics and Molecular Biology; Immunology and Microbiology; Pharmacology, Toxicology and Pharmaceutics; and Chemistry, covering the period from 1972 to December 5, 2024." |
|
Methodology: Change to First, the scientometrics |
We agreed with this comment and changes were made. |
|
Results: Figure 3: Identify each graph with a letter and provide a more complete legend. |
Figures and legends were changed. |
|
Results: the same for Figure 4 ; Figure 4 could be divided by letters; A,B and C and a more complete legend. |
|
|
Results: check this [e.g., 22,23] (e.g.,). (e.g.,) |
e.g. were deleted. |
|
Results: Cluster 2 investigates - What is the colour of this cluster? |
Colors were added. |
|
The approach used by the authors to comment the studies by author (Table 3) does not seem ethical. Microbiome/microbiota science is not individual. There is a principal investigator but the studies report data worked by a team. I suggest to revise this approach (as principal investigator instead of principal author). |
We agreed with the comments and changes were made. |
|
Page 6, where principal author is a review in which about change to principal investigator is a review about |
We agreed with the comments and changes were made. |
|
Page 7, Change Professor Heping Zhang, to Researcher Heping Zhang, |
|
|
Page 7, Change Dr. Zhang to Zhang et al |
|
|
Page 7, as above change Professor Jiachao Zhang to Researcher (as above) |
|
|
Page 7. Change Dr. Jiachao Zhang’s recent work to in recent studies of J. Zhang’s team |
|
|
Page 8, change microeukaryotes to eukaryotes (in the sentence you have microorganisms, so the designation microeukarayotes is not correct). |
|
|
Page 8, the definition of microbiome provide here comes from the reference of Berg et al. 2020 (Berg, G., Rybakova, D., Fischer, D. et al. Microbiome definition re-visited: old concepts and new challenges. Microbiome 8, 103 (2020). https://doi.org/10.1186/s40168-020-00875-0), please cite it instead of reference 36. |
Thank you for the suggestion. The reference was added. |
|
Page 9 Verrumicrobia is presently Cite for the new phylum taxonomy the bellow reference: Oren A, Garrity GM. Valid publication of the names of forty-two phyla of prokaryotes. Int J Syst Evol Microbiol. 2021 Oct;71(10). doi: 10.1099/ijsem.0.005056. |
We agreed with this comment and changes were made. |
|
Page 9, Change húesped to host |
We agreed with this comment and changes were made. |
|
Page 10 Figure 6 is adapted from reference 61? Provide a better legend and the authorization for the use. |
The image was replaced with our own version, based on the information from the cited reference. |
|
Page 10, Change The intestine to The gastrointestinal tract . (The intestine does not includes the stomach!) |
We agreed with this comment and changes were made. |
|
Page 11, LMH in full |
Changed. |
|
Page 11, correction required for 1x105 to 1x107. |
Changed. |
|
Page 11, Also indicate here: Firmicutes (~65%), Bacteroidetes (~25%) Actinobacteria (~5%) and Proteobacteria (~5%), the members of Proteobacteria, Fusobacteria, Cyanobacteria and Verrucomicrobia " the new taxonomy |
We agreed with this comment and changes were made. |
|
Page 12, Bacteroidetes : indicate the new taxonomy |
Changed. |
|
Page 12, change a prime example is Bacteroides thetaiotaomicron to a prime example is thetaiotaomicron |
Changed. |
|
Page 12 Again for Firmicutes indicate the new taxonomy. Here and all over the manuscript |
Changed. |
|
Page 12, what this means generate high fermentation ? |
We had a translation issue. We reviewed the idea again and it was corrected. Lines 425-426. |
|
Page 12 change short-chain fatty acids (SCFA) acetate to short-chain fatty acids (SCFA), namely acetate |
Changed. |
|
Rewrite the sentence: Proteins: Amino acids (aa) are the functional units of proteins. These proteins ....( this is not appropriate) |
The sentence was rewritten |
|
Page 12, Please revise this: These amino acids are not an energy source for bacteria, as they lack all the necessary enzymes for conversion. See the aminoacid fermentation by Clostridium species. where aminoacids are used as electron donors and acceptors (tryptophan and tyrosine can be both). See Stickland fermentation. |
The sentence was deleted. |
|
Page 13, Modify this Lipids: Lipid …This is not an appropriate format for starting a sentence. |
The sentence was rewritten. |
|
Page 14 A more complete of legend of Figure 6 is required. |
We agreed with this comment and changes were made. |
|
Page 14, change Gut Microbiome to gut microbiome |
Changed. |
|
Page 15, change Immune System to immune system |
|
|
Page 15, Change Microbial metabolites can inhibit to SCFAs can inhibit |
|
|
Page 16, Againg change Gut Microbiome to gut microbiome |
|
|
Page 16, change Gut-Lung Axis to gut-lung axis |
|
|
Page 16 Change Cardiovascular Diseases to cardiovascular diseases |
|
|
Page 16, Again change Gut Microbiome to gut microbiome |
|
|
Page 16, change Gut Microbiota to gut microbiota |
|
|
Page 17, if the intention is to use of the genus names change to Bifidobacterium and Lactobacillus if the bacterial group change to bifidobacteria and lactobacilli. |
We remain the bacterial group. Italics and capital letters were modified. |
|
Page 17, change gut flora to gut microbiota |
We agreed with the comments and changes were made. |
|
References, put the reference 3 in the required format. |
Thank you once again for your time and insightful feedback.
Reviewer 2 Report
Comments and Suggestions for Authors
Review report:
I would like to congratulate the authors on this comprehensive manuscript and presenting the latest findings that deepen our understanding of the role of the human microbiome in human health. In addition, the manuscript explores new opportunities for the developing treatments and emphasizes the importance of maintaining a balanced microbiome for promoting health and preventing diseases.
The manuscript is written precisely and clearly, with enough emphasis on the issues it deals with. However, I would suggest a few additions to the sections "Metabolism of macronutrients" and "Impact on human health" and a rewriting of the Conclusion.
- The sections mentioned would benefit from the addition of the names of bacterial species or genera that are associated with the beneficial or harmful effects discussed by the authors.
- In its current form, the Conclusion closely resembles the Introduction. I recommend rewriting the Conclusion so that it aligns with the study's objectives and reflects the key findings presented in the manuscript.
Author Response
Dear Reviewer,
Thank you so much for the time to review this manuscript. We sincerely appreciate your valuable comments. Please find the detailed responses below.
Comments 1: I would suggest a few additions to the sections "Metabolism of macronutrients" and "Impact on human health" and a rewriting of the Conclusion.
The sections mentioned would benefit from the addition of the names of bacterial species or genera that are associated with the beneficial or harmful effects discussed by the authors.
Response 1: We agreed with your comment and we added the genera and bacterial species in lines: 416,-422, 495-501, and 584-589.
Comments 2: In its current form, the Conclusion closely resembles the Introduction. I recommend rewriting the Conclusion so that it aligns with the study's objectives and reflects the key findings presented in the manuscript.
Response 2: We appreciate the opportunity to further elaborate on the conclusions and specify those findings we consider significant for the academic community. Consequently, we rewrite the conclusions from line 682 onward.